# The Epidemiologic Transition in French Guiana: Secular Trends and Setbacks, and Comparisons with Continental France and South American Countries

**DOI:** 10.3390/tropicalmed8040219

**Published:** 2023-04-08

**Authors:** Mathieu Nacher, Célia Basurko, Maylis Douine, Yann Lambert, Najeh Hcini, Narcisse Elenga, Paul Le Turnier, Loïc Epelboin, Félix Djossou, Pierre Couppié, Bertrand de Toffol, Kinan Drak Alsibai, Nadia Sabbah, Antoine Adenis

**Affiliations:** 1Centre Hospitalier de Cayenne, CIC INSERM 1424, 97300 Cayenne, French Guiana; 2Amazonian Infrastructures for Population Health, 97300 Cayenne, French Guiana; 3Département Formation Recherche, Université de Guyane, Campus de Troubiran, 97300 Cayenne, French Guiana; 4Centre Hospitalier de l’Ouest Guyanais, 97320 Saint Laurent du Maroni, French Guiana; 5Centre Hospitalier de Cayenne, Pédiatrie, 97300 Cayenne, French Guiana; 6Centre Hospitalier de Cayenne, Service des Maladies Infectieuses et Tropicales, 97300 Cayenne, French Guiana; 7Centre Hospitalier de Cayenne, Service de Dermatologie, 97300 Cayenne, French Guiana; 8Centre Hospitalier de Cayenne, Service de Neurologie, 97300 Cayenne, French Guiana; 9Centre Hospitalier de Cayenne, Service d’Endocrinologie Diabétologie, 97300 Cayenne, French Guiana

**Keywords:** epidemiologic transition, infectious diseases, non communicable diseases, tropical, life expectancy, premature mortality

## Abstract

There are great variations between population subgroups, notably in poorer countries, leading to substantial inconsistencies with those predicted by the classical epidemiologic transition theory. In this context, using public data, we aimed to determine how the singular case of French Guiana fit and transitioned in the epidemiologic transition framework. The data show a gradual decline in infant mortality to values above 8 per 1000 live births. Premature mortality rates were greater but declined more rapidly in French Guiana than in mainland France until 2017 when they reascended in a context of political turmoil followed by the COVID-19 pandemic and strong reluctance to get vaccinated. Although infections were a more frequent cause of death in French Guiana, there is a marked decline and circulatory and metabolic causes are major causes of premature death. Fertility rates remain high (>3 live births per woman), and the age structure of the population is still pyramid-shaped. The singularities of French Guiana (rich country, universal health system, widespread poverty) explain why its transition does not fit neatly within the usual stages of transition. Beyond gradual improvements in secular trends, the data also suggest that political turmoil and fake news may have detrimentally affected mortality in French Guiana and reversed improving trends.

## 1. Introduction

The epidemiologic transition perspective describes the changing patterns of population distribution and those of the main causes of death, mortality, fertility, and life expectancy [1,2]. The transition entails two major changes: changes in mortality patterns, including increasing life expectancy and ranking of the main causes of death, and changes in population growth and composition. However, all countries do not follow the same sequence of transitions, and the universality of the theory has hence been questioned. There are great variations in mortality trends between population subgroups, notably in poorer countries, leading to substantial inconsistencies with those predicted by the classical epidemiologic transition theory. Although the theory has been criticized, revised, and falls short in terms of explaining or predicting, it is still a relevant way of understanding the relation between disease, mortality patterns, and population dynamics. However, there is often a lack of reliable mortality data which makes it difficult to test whether the theory can be generalized globally [1,2].

French Guiana is a French overseas territory located on the South American continent, between Brazil and Suriname. At EUR 15,260 per capita, French Guiana has the highest gross national product (GNP) per capita in Latin America and attracts many immigrants seeking a better life [3]. Thus, 29% of the population and nearly half of the adults are of foreign origin. Furthermore, over half of the population in French Guiana lives below the French poverty threshold of less than EUR 1010 per month [4]. French Guiana has the highest fertility rate in Latin America (3.56 children per woman) and sustained immigration; hence, its small population is expected to double in about fifteen years. [5,6] Despite these conditions reminiscent of low- and middle-income countries, French Guiana has a modern French universal health care system and, by far, the highest health expenditure per capita in Latin America (double that of Chile, the second highest) [7]. The communes in the interior of French Guiana—mostly surrounded by Amazonian Forest—are inaccessible by road. The populations that reside there are essentially Amerindians or Maroons. The prevention and care offered in the remote areas is provided by a network of 17 health centers. French Guiana thus offers a complex mix, with wealth and poverty, a universal health system, free education, and free mother and child care. In these remote areas, tropical infectious diseases remain prevalent and are the focus of most medical research. [8] Hence, because of this research bias, French Guiana is still viewed as a land where infectious diseases are the major health burden. Although in recent years research on chronic diseases has started to increase, there has been no systematic effort to put the evolution of the burden of disease in perspective in this singular territory. French Guiana is on the South American continent, and health outcomes partly result from the ecosystem of pathogens, the environment, the cultures, and the ethnic and genetic make-up of local populations. However, the French universal health system and social provisions to alleviate health inequalities are also important determinants of health outcomes and life statistics in French Guiana. It is therefore meaningful to benchmark French Guiana against continental France in order to pinpoint territorial disparities regarding health outcomes. In this singular context, we aimed to compile temporal data on mortality, the main causes of death, birth rates, and life expectancy at birth to determine how French Guiana fit and transitioned in the epidemiologic transition framework; we also aimed to compare it with its administrative center, mainland France, and with other countries in South America.

## 2. Methods

We queried the National Institute for Statistics and Economics Studies (INSEE) databases to obtain demographic statistics, life expectancy, and mortality statistics up to 2021. [6] Statistics for causes of death for French Guiana and mainland France were obtained through the CEPIDC, which compiles information from death certificates for the National Institute for Health and Medical Research (INSERM) [9]. The standardized death rates provided by the CepiDC used the 2006 census French population. The CepiDC data on causes of death take longer process than INSEE data and were only available up to 2017. Data for Latin America were obtained from ourworldindata.org [5]. Three types of visual comparisons were made: first, between French Guiana and mainland France because both share the same health system and same health rights; comparisons between French Guiana and South American countries because they share the same continent; and finally, temporal trends were plotted. Preventable causes of death are causes of death under the age of 65 linked to risky behavioral factors, in line with the criteria used in France (AIDS, cancers of the upper aerodigestive tract, cancers of the trachea, bronchus and lung, alcoholic psychoses, cirrhosis, traffic accidents, accidental falls, and suicides) [10].

Data were plotted using STATA 16 (Stata Corp, College Station, TX, USA) and Microsoft Excel (Microsoft, Redmond, WA, USA) for the treemap.

## 3. Results

### 3.1. Mortality Statistics

Different mortality indicators are used to approach the situation in French Guiana and to contrast it with mainland France.

### 3.2. Evolution of the Standardized Death Rate, Premature Mortality, and over 64 Mortality

Figure 1A shows the standardized (2006 French population census) death rate gradually declined until 2016 both in French Guiana and mainland France (*p* < 0.001) but remained higher in French Guiana. There was an inflection starting in 2017 (the year of a month-long blockade of major roads followed by prolonged hospital strikes) with an increase in French Guiana also shown in premature deaths (Figure 1B). After this inflection, the substantial impact of the 2020–2021 period of the COVID-19 pandemic is captured by Figure 1B,C. The oscillations in Figure 1A,C reflect the random fluctuations from one year to another due to the small population of French Guiana.

Figure 1B shows there was a gradual decline of premature mortality (<65 years) until 2016 both in French Guiana and mainland France. The difference between French Guiana and mainland France gradually narrowed until 2017, the year of a month-long blockade of major roads and prolonged strikes in French Guiana, concomitant with a reversal of the decline in French Guiana prolonged by the consequences of COVID-19 pandemic, which were apparently worse in French Guiana than in mainland France.

Overall, Figure 1C shows that among those aged over 64 years there was no difference between French Guiana and mainland France, until 2021, when successive waves of COVID-19 affected French Guiana, the South American territory with the lowest vaccination rate.

### 3.3. Evolution of Under-5 Mortality

Figure 2 shows that for the past three decades, under-5 mortality (shown using a base-10 logarithmic scale for clarity) has declined (0.97% in 2021) but remains higher than Chile (0.56% in 2021) and Uruguay (0.56% in 2021), for South America and mainland France (0.42% in 2021). The difference with France has narrowed, but the rate in French Guiana remains over twice that of France. Similarly, for infant mortality, the rate in French Guiana (0.9%) is over twice that of France (0.35%).

### 3.4. Main Causes of Early Death

The treemaps in Figure 3 and Figure 4 show the breakdown of causes of death <65 years for 2001–2017 in French Guiana and mainland France, respectively. French Guiana distinguishes itself from mainland France by the greater contribution of circulatory diseases; endocrine, nutritional, and metabolic diseases; infectious diseases; and diseases linked to pregnancy (perinatal, chromosomal, and congenital diseases) and the lower contribution of tumors to overall mortality. The frequency of poorly defined causes is linked to the lack of information in the death certificate when there is a lack of medical history.

When zooming in on the external causes of death among persons aged <65 years in French Guiana, the following causes were significantly greater in French Guiana than mainland France: accidents and transport accidents, 52.7% versus 47.6%, respectively; homicides, 7.4% versus 1.3%, respectively; and drowning, 7.2% versus 2.3%, respectively. By contrast, among external causes, suicide was more frequent in mainland France (28.7%) than in French Guiana (10.4%).

### 3.5. Evolution of the Main Causes of Death

In French Guiana, although standardized death rates from circulatory diseases seem to follow the marked decline observed in mainland France, the latter period seems to show a rebound. Tumors as causes of death fluctuate but seem stable and below rates in mainland France, which show a steady decline. Standardized death rates from endocrine, nutritional, and metabolic causes; external causes; and infectious diseases are generally greater than in mainland France but also show a gradual temporal decline. Figure 5 and Figure 6 show the evolution of major causes of death in French Guiana using age-standardized rates in comparison with mainland France. The curves from French Guiana generally fluctuate because of lower numbers.

### 3.6. Evolution of Infections as a Cause of Premature Death

When zooming in on infectious diseases, major gains were seen for HIV both in French Guiana and mainland France (Figure 7). Although the decline was steeper in French Guiana than in France, the death rate from HIV/AIDS remains greater in French Guiana. Regarding infections excluding HIV/AIDS, tuberculosis, and hepatitis, there was also a decline both in mainland France and in French Guiana, but the decline was more pronounced in French Guiana where after 2010 levels became lower than in mainland France (Figure 7).

### 3.7. Life Expectancy

#### 3.7.1. Life Expectancy at Birth

Life expectancy at birth increased until 2020, but there is a steady 2–3-year gap for females in French Guiana and a 2-year gap for males (Figure 8a). However, in 2021, COVID-19 and low vaccine uptake led to a sharp decline of life expectancy at birth in French Guiana relative to mainland France: 5.5 years difference for females and 6.7 years for males.

#### 3.7.2. Life Expectancy at 20, 40, and 60 Years

Excluding 2021, when looking at life expectancy at ages 20, 40, and 60 years (Figure 8b–d), it appears that in males at 40 and 60 years, life expectancy is similar to that in mainland France. At 20, however, it is lower than in mainland France. By contrast, for females, life expectancy at 20, 40, and 60 is consistently lower than in mainland France.

#### 3.7.3. Gains

On a greater timescale, between 1951 and 2015, French Guiana gained over 27 years of life expectancy, the greatest gain in South America, and over the twenty years before 2015, French Guiana gained over 6 years in life expectancy, the greatest recent gains (pre-COVID-19) being observed in Brazil (Appendix A).

#### 3.7.4. Population Growth and Structure

Population growth in French Guiana remains among the fastest in Latin America. It is largely driven by a very high birth rate and a favorable natural balance (Appendix A). The fertility rate is around 3.5 children per woman, while the second highest in South America is Bolivia (2.65). The age structure of the population illustrates its youth (median age 24 years) with a broad base pyramid-like structure (Figure 9) showing the marks of substantial emigration among young adults and a very small population over 65 years. Appendix A shows the age pyramid of France for contrast.

## 4. Discussion

Here, we show how different French Guiana is from mainland France in demographic and epidemiological terms. The standardized death rate remains greater in French Guiana than in mainland France. A number of findings are evocative of low- and middle-income countries with a greater burden of infections and obstetrical/perinatal causes in French Guiana than in mainland France: Child and infant mortality are still substantially greater in French Guiana than in mainland France. The high infant mortality presumably reflects reality given poor pregnancy follow-up and the importance of preterm delivery in French Guiana [11]; however, there have been suspicions that cultural opposition to interruptions of non-viable pregnancies results in the delivery of live births that die soon thereafter [12]. The main causes of premature death are different, with more infections (arboviral epidemics) and more deaths related to perinatal causes in French Guiana than in mainland France. Pregnant women are also exposed to saturnism, anemia, obesity, and consanguinity, which can play a role in increasing risk of mortality. For various reasons, pregnant women from neighboring countries often prefer to give birth in French Guiana [13]. These pregnancies are often unattended and at greater risk of perinatal death. It was surprising to see infections other than HIV/tuberculosis or hepatitis decline in French Guiana to lower values than in mainland France. We suspect this may have been related to the broader base in the age pyramid in French Guiana than in mainland France, hence greater resistance to infection.

Although cancers are overall less frequent, infection-related cancers (notably cervix and stomach) are more frequent in French Guiana than in mainland France [14,15,16]. However, the transition towards chronic diseases is well underway with a decline of infectious diseases—not only of HIV—and a greater burden of circulatory and metabolic diseases in French Guiana than in mainland France.

In 2017, political movements (notably against the perceived poor state of health care in French Guiana) prompted a month-long blockade of all major roads in French Guiana followed by 74 days of strike at Cayenne Hospital [17]. This disrupted services and led many patients with chronic diseases to interrupt care, or patients with acute life-threatening conditions to refrain from going to the hospital for urgent care. These prolonged events are contemporaneous with, and perhaps explain, the reascension of premature mortality after years of regular progress and a narrowing of the difference with the rate in mainland France; this inflection point was followed by the impact of the pandemic, which may have further increased premature mortality.

In terms of life expectancy at birth, French Guiana has had a steady 2–3-year gap with mainland France, much of which results from the premature causes of death, notably in young age groups: perinatal issues due to poor pregnancy follow-up, or accidents. This life expectancy deficit with mainland France is often portrayed as an illustration of the failure of the hospital system resulting in a greater risk of dying in French Guiana [18]. However, the comparative analysis of life expectancy at 20, 40, and 60 years reveals interesting sex differences. First, among males, there is no obvious difference in life expectancy at 40 and 60 years between French Guiana and mainland France. The main epidemiological risk factors differ—smoking and alcohol consumption are less frequent in French Guiana than in mainland France, whereas high blood pressure and obesity are more frequent—so the health system may be facing a different mix of diseases. Nevertheless, this suggests that, despite the widespread poverty at ages where serious pathologies become increasingly frequent, notably in males, there is no obvious detriment for living—and being treated—in French Guiana relative to mainland France, a counterintuitive observation. As described previously, according to the definition of avoidable death mortality, between 33.9% (French definition) and 52% (UK definition) of the causes of premature death are sensitive to primary care (in other words, preventable) [7].

Among males, life expectancy at 20 years of age is, however, consistently lower than in mainland France. This perhaps reflects the greater importance of violent deaths from external causes in French Guiana. Among females, by contrast, life expectancy at 20, 40, and 60 years is consistently lower in French Guiana than in mainland France. Given the importance of metabolic and circulatory causes of death, this may reflect the fact that obesity (BMI > 30) in French Guiana is very prevalent, especially in women (23% vs. 15% among men) [19,20]. In 2021, as French Guiana was hit by prolonged waves due to different COVID-19 variants, life expectancy dropped by 4–5 years relative to mainland France (which was also affected by COVID-19). This is presumably explained by vaccination rates that were much lower (33% received at least one dose) than those achieved in mainland France (82% received at least one dose) with widespread opposition to vaccines in French Guiana, actively encouraged by local political leaders and unions [21]. A similar drop in life expectancy was observed in Martinique and Guadeloupe—two territories with very low vaccination uptake—which gives further credence to this hypothesis. A complementary explanation for the 2021 drop in life expectancy is that the disruption caused by COVID-19 in terms of health care services for non-COVID-19 issues, the impoverishment of the most miserable [18], and difficulties in circulation may have led to increased mortality from other causes.

In terms of population growth and structure, French Guiana is clearly an outlier in South America with the highest fertility rate and the population structure of a low/middle income country, despite having a life expectancy that is closer to those of OECD countries. A major premise of the epidemiologic transition theory [1,2] is that a long-term shift in mortality and disease patterns takes place with pandemics, and infectious diseases are gradually replaced by chronic and degenerative diseases. The most profound changes in health and disease patterns during the epidemiologic transition occur among children and young women, with declining infant and maternal mortality. Declining mortality is typically followed by declining fertility, and thus lower birth rates, lower death rates, and higher life expectancy have profound demographic consequences. Here, however, the picture is incomplete at least in Western French Guiana where Maroons still have very high fertility rates [22], and the expected demographic changes are only beginning [23]. The complex ethnocultural and linguistic make up of French Guiana presumably leads to substantial differences in life statistics and causes of death, but the data used here are aggregated data that do not allow such aspects to be observed.

The epidemiologic transition changes in health and disease patterns are closely associated with the demographic and socioeconomic transitions of modernization. Thus, improved socioeconomic status leads to better nutrition and sanitation, and further gains in morbidity and mortality. These propositions result in five stages of transition [24]. First, the age of pestilence and famine, with high and fluctuating mortality from epidemics, famines, war, and poverty. This stage entails high crude death rate, high fertility rate, and low life expectancy at birth and slow population growth. The main causes of death are infections, especially among children and women. The second stage is the age of receding pandemics, where mortality rates decline, later followed by decreasing fertility (mostly during the third stage) and increasing life expectancy at birth. This stage is reached through sanitation improvements, control of infectious disease outbreaks, and medical progress. Although in this second stage infectious diseases remain major causes of death, the burden of non-communicable diseases gradually increases. The third stage is the age of degenerative and man-made disease, which entails decreasing and stable low mortality and increasing life expectancy at birth, a rapidly declining birth rate, and an aging population. In this stage, non-communicable diseases are the main causes of death, with many deaths attributable to cardiac and cerebrovascular diseases, metabolic diseases, cancers, injuries, and stress-related diseases. A fourth stage describes the age of declining cerebrovascular mortality, where improved medical care and lifestyle modifications cause the mortality from cardiovascular diseases to decline and stabilize. Finally, a fifth stage is characterized by the emergence of new diseases (HIV/AIDS…) and the re-emergence of old infectious diseases [5].

Our data show that French Guiana does not fit neatly in the above stages. In epidemiological terms it would fit in between the third and fourth stages, with a decline of the burden of infectious diseases and child mortality, but an unusual demographic pattern—the curves from the standard theory usually show a decline of death rates marking the second stage of transition and a rapid decline in birth rates during the third stage—with a sustained birth rate despite lower mortality. This may result from the unusual mix of poor populations living in a rich country with a universal health system and free education.

The present study has a number of limitations. The correlations between the different indicators tell an interesting story, but they are not necessarily causal relations. The INSEE data for the last 3 years are provisional and may be slightly refined. Given the small population size in French Guiana, random variations led to jagged temporal trends, requiring trend lines to neutralize irregularities. In French Guiana, over half of the population lives under the poverty level. It is well known that, for infectious or non-communicable diseases, the most deprived patients consult later, with more severe presentations at a younger age [4,25,26,27]. For breast cancer survival, for instance, we have shown that crude survival was lower than in mainland France but that, when only considering French-Guiana-born women, there was no longer any difference with mainland France. Furthermore, nearly 30% of the population and half of adults are immigrants who often have greater fertility rates. Hence, while natality among French nationals was stable, it increased in foreigners living in French Guiana, with three out of five babies born from a foreign mother and two out of five from a foreign father [28]. Therefore, the mortality, fertility, and life expectancy curves shown here aggregate very different situations in a very heterogeneous population, which presumably explains why the singular profile of French Guiana does not fit neatly in the usual dynamics of epidemiological transition. Finally, the comparison to other countries that can be fit in per capita income as poor < developing/transitioning < developed may be questionable since French Guiana cannot be easily categorized in this manner.

In conclusion, we show the rapid changes of mortality, causes of death, and demographic parameters in French Guiana. This complex French territory is administratively identical to any French “departement” but is at a different stage from mainland France in the epidemiologic transition (the shift in disease patterns) and the demographic transition (the shift in vital statistics) frameworks [24]. The demographic and epidemiological singularities of the territory explain why its transition does not fit neatly within the usual stages of transition—demographic changes seeming slower than the epidemiologic transition. In the highly centralized French system, it is important to incorporate these specificities in strategic priorities. Beyond gradual improvements in secular trends, the data also suggest that political turmoil, blockades, and vaccine hesitancy may have detrimentally affected mortality in French Guiana and reversed improving trends.

## Figures and Tables

**Figure 1 tropicalmed-08-00219-f001:**
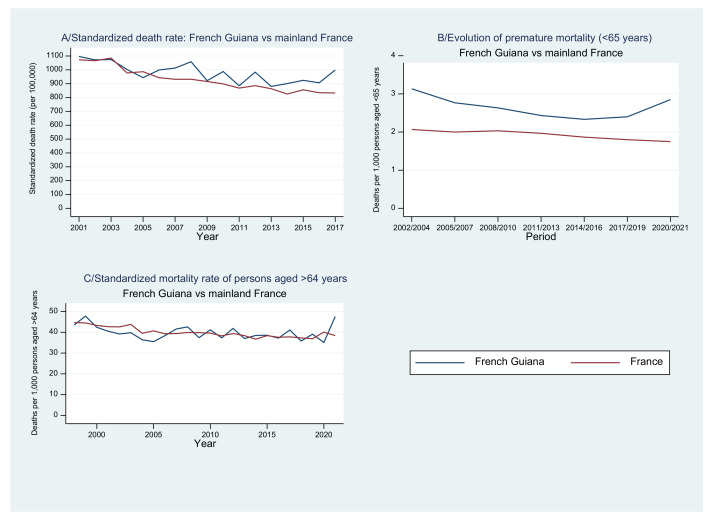
(**A**–**C**): Evolution of mortality indicators in French Guiana and mainland France: 2001–2021.

**Figure 2 tropicalmed-08-00219-f002:**
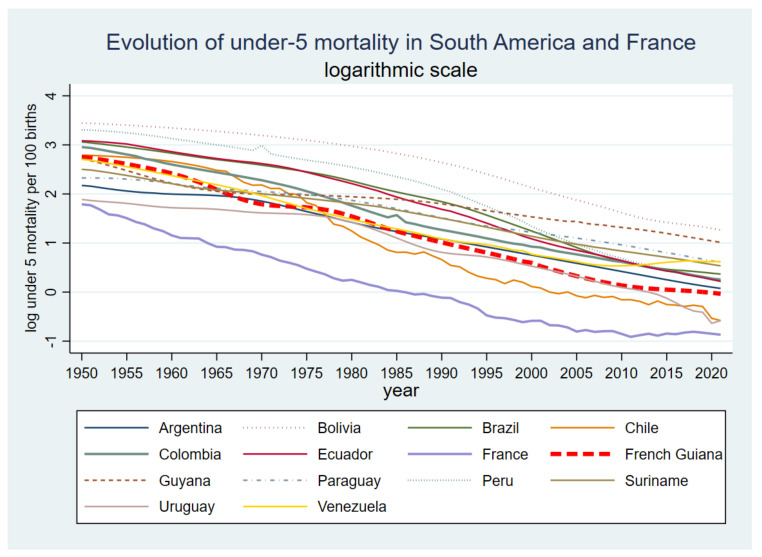
Evolution of under-5 mortality in South America and France (log10 scale).

**Figure 3 tropicalmed-08-00219-f003:**
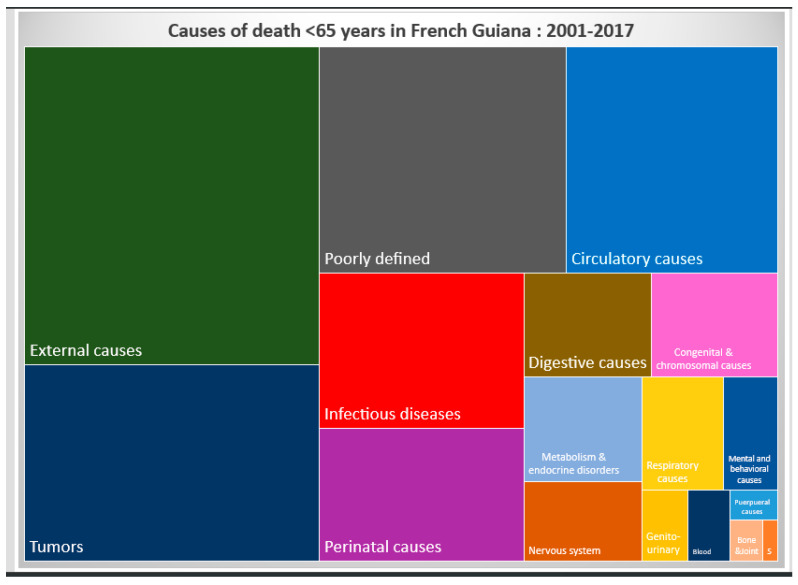
Treemap of the causes of premature deaths in French Guiana.

**Figure 4 tropicalmed-08-00219-f004:**
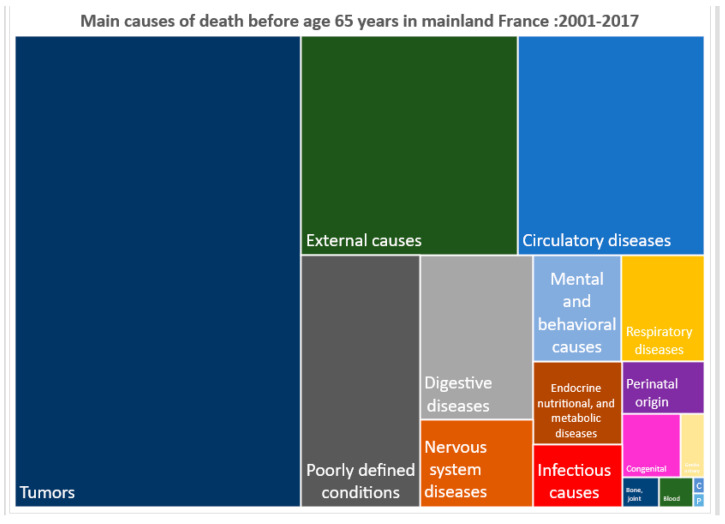
Treemap of the causes of premature deaths in mainland France.

**Figure 5 tropicalmed-08-00219-f005:**
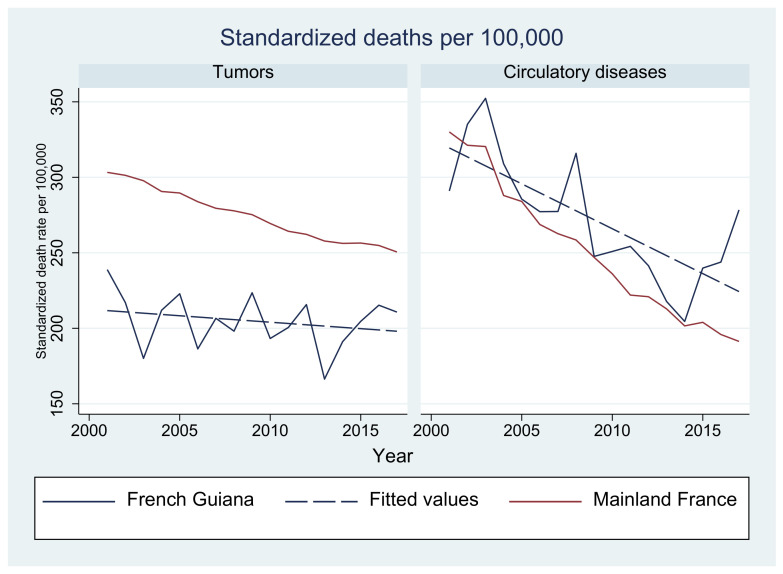
Evolution of the standardized death rate for tumors and circulatory diseases: French Guiana vs. mainland France.

**Figure 6 tropicalmed-08-00219-f006:**
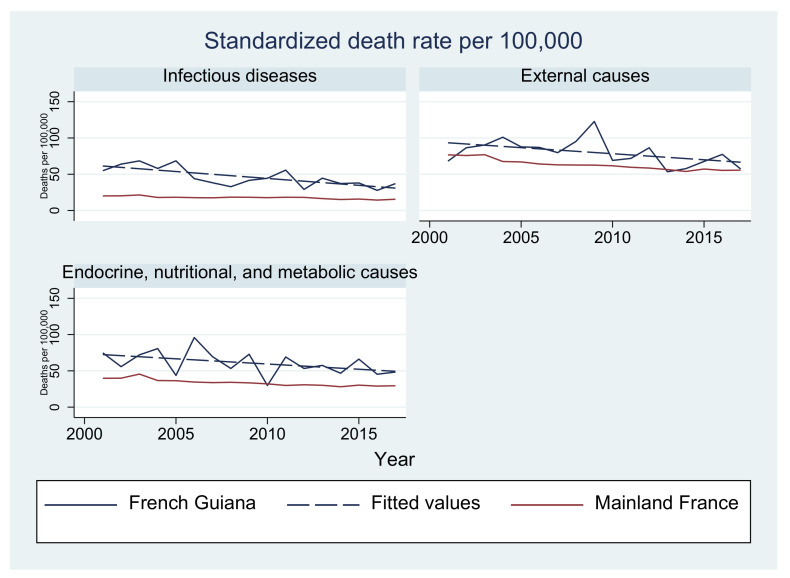
Evolution of the standardized death rate for infectious diseases, external causes, and endocrine, nutritional, and metabolic diseases: French Guiana vs. mainland France.

**Figure 7 tropicalmed-08-00219-f007:**
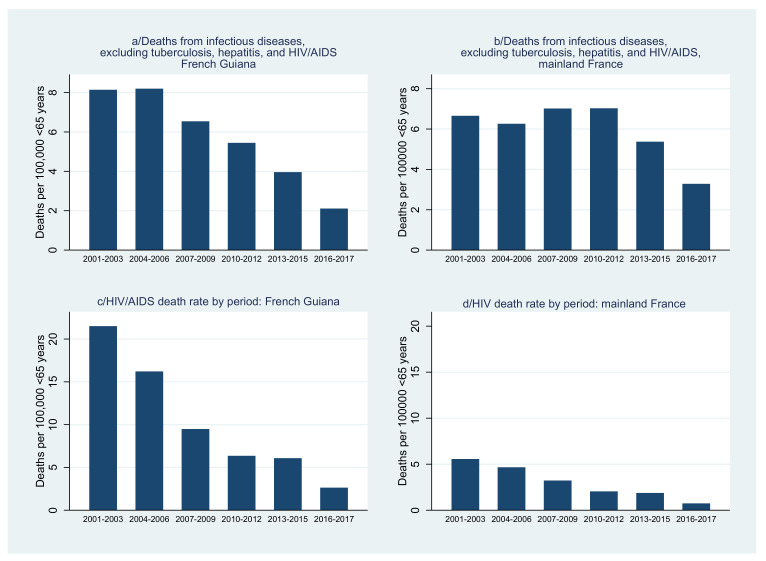
Evolution of the standardized death rate for infectious diseases other than tuberculosis, hepatitis, and HIV/AIDS (French Guiana (**a**) vs. mainland France (**b**)), and for HIV/AIDS (French Guiana (**c**) vs. mainland France (**d**)).

**Figure 8 tropicalmed-08-00219-f008:**
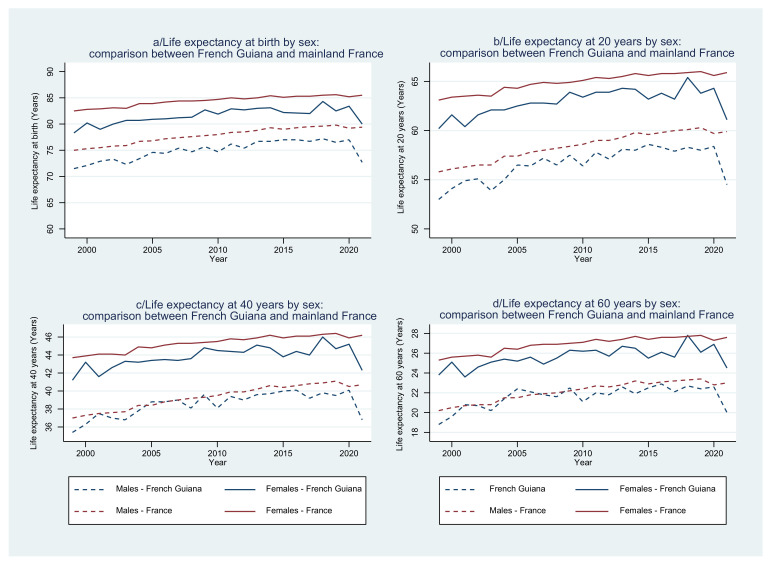
(**a**–**d**): Evolution of life expectancy by sex at birth (**a**), at 20 years (**b**), at 40 years (**c**), and at 60 years (**d**): French Guiana vs. mainland France. The y axis represents the number of years one could expect to live at birth, 20 years, 40 years, and 60 years.

**Figure 9 tropicalmed-08-00219-f009:**
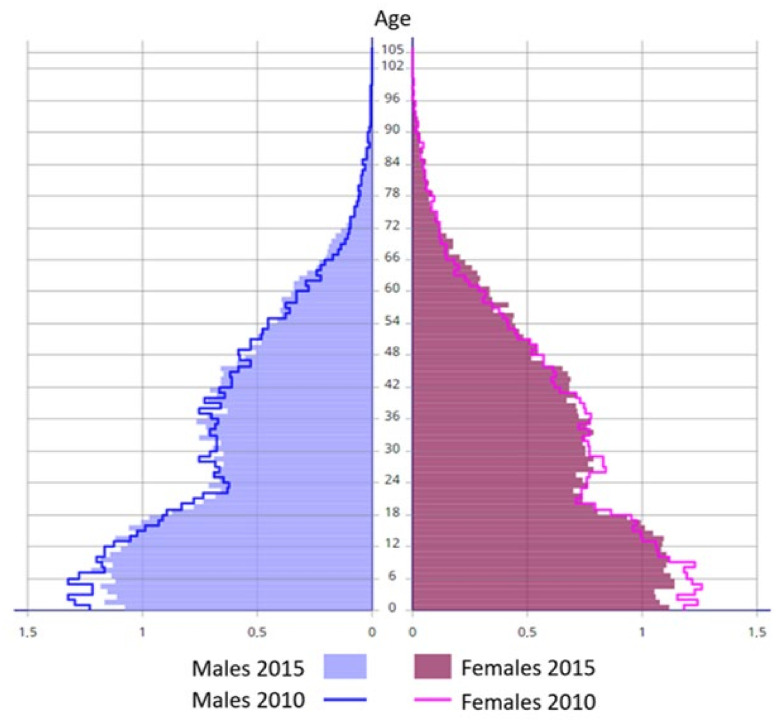
Age pyramid of French Guiana. Figure legend. The y axis represents age, and the x axis represents population by 1-year age group in thousands. The lines represent values and shape for 2010, whereas purple and salmon colors represent values and shape for 2015.

## Data Availability

The data is already publicly available.

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
