# Peer review of "The Epidemiologic Transition in French Guiana: Secular Trends and Setbacks, and Comparisons with Continental France and South American Countries"

_tropicalmed, 2023, doi:10.3390/tropicalmed8040219_

Round 1
Author Response
Dear Reviewer, we are very grateful for your in-depth analysis and suggestions. It has been very helpful to help us see the manuscript in a new light. We have inserted answers to your points hereunder.
We hope you will find it has sufficiently improved.
Sincerely
The Authors
Review of „The epidemiologic transition in French Guiana: secular trends 2 and setbacks, and comparisons with continental France” (tropicalmed-2285797) The authors submitted a report of the status and the progression of the classical epidemiological transition theory of French Guiana by determining the singular case for French Guiana can be fitted within this framework. The authors move on to compare the data to mainland France and other South American countries. The article is well written and has a good size. It further addresses important aspects that hallmark epidemiological transition and hence evolution. The data has been raised carefully and a praisable effort has been undertaken to gather missing data or data that is in general difficult to acquire because it hasn’t been documented routinely all the time. In this context the collected date is very much presentable and by itself interesting to the scientific community.
=>Thank you
However, I did have some difficulties with scope and aim of this article and how the authors decided to present such data. It is not entirely clear to my why the authors chose to compare the data i) to mainland France and ii) to some South American countries. Whereas there might be a valid argument for the second point I fail to see the greater value of comparison to mainland France.
The data can certainly be compared that much is undisputable but in my opinion the authors fail to state why they included certain countries for comparison and left out others. The mere fact that French Guiana is an overseas French territory alone may be comprehensible for comparison on a political scale but it alone does not justify selection in a life science, medical and public health field and at least should be explained in more detail, which I strongly suggest to the authors to do.
=>indeed we are on the South American continent and health outcomes partly result from the ecosystem of pathogens, the environment, the cultures, and the ethnic and genetic make-up of local populations. However, the French universal health system and social provisions to alleviate health inequalities are also important determinants of health outcomes and life statistics. It is always useful to benchmark against continental France in order to pinpoint singularities. Furthermore, the national information systems, statistics allow us to make reliable comparisons for different indicators. This is why we compare with mainland France. Although we would be eager to compare all indicators with South American countries, we do not have access to most indicators to do so. We tried to better explain this in the introduction.
Additionally, the situation in French Guiana is quite distinguished from other South American countries as it is from mainland France. It therefore functions more as an example for an unusual classical epidemiological transition and possibly cannot be fitted in the suggested framework properly.
=>indeed we also think it is unusual and for this reason we think it is interesting
It may not even be a proper case for comparison to other countries that can be fitted in per capita income as poor < developing/transitioning < developed since French Guiana cannot be easily categorized in this manner. But this hasn’t been discussed or at least not extensively enough for the reader. Hence, I wonder what the exact gain of knowledge we can get from the chosen comparison and the study design in its current state. It would be interesting to receive the author’s perspective on this and I would like to ask that the raised points should be addressed and included in the manuscript.
=>thank you, we agree French Guiana is not easily categorized and thus comparisons with different countries with low or medium ressources, or high ressources are difficult we have added this point in the limitations section.
Further, following question should also be addressed in more detail: Why was the acquired data for French Guiana not compared to other French overseas territories with a similar population size and a similar health care / public health structure and possibly even epidemiological situation such as Reunion, Guadeloupe/Martinique, New Caledonia and so on? I believe such comparison in the manner already performed by the authors for French Guiana would be highly anticipated and a strong collection for the scientific community. I suggest to include this data.
=>thank you for this suggestion. The problem would be that including all these complex territories would require detailed social, epidemiological, and demographic prerequisites to enlighten the comparisons. The present paper already has many figures and supplementary figures and the number would further increase. However, it is a great idea for a specific paper, perhaps less centered on French Guiana, comparing all these territories which are often seen as somewhat interchangeable “tropical islands” from mainland France.
Following other points I would like to be addressed in line order. 1. Line 1-3: The title states comparison to mainland France but later in the manuscript comparison to other South American nations is also referenced. The title or the data should be in line with what the authors mean to present. 2.
=>OK, we have modified the title to encompass the full scale of comparisons
Lines 31-33: There are some statements of turmoil within the manuscript but it hardly addresses the very valid issue of “fake news”. This is taken up at the very and of the discussion but is not part of intensive investigation in the study. The points may be raised but the author’s claim that their data suggests that these points affect mortality detrimentally. If so then this must be discussed in more detail and data that supports this directly should be added.
=>the paper deals with secular trends and setbacks and indeed fake news were quite prevalent in French Guiana. However, apart from press articles emphasizing the problem and our own witnessing of what was going around we do not have any specific data on fake news (surveys). We have refrained to open a new hot topic in the conclusions section which was a mistake. We thus rephrased to vaccine hesitancy which was mentioned before in the text and does not raise a new topic late in the manuscript.
- L99 - 106: I would challenge the gradual decline in 1A for French Guiana. It is more an up and down and this may well be attributable to smaller data sets but the conclusion of a gradual decline is not supported as it is for mainland France.
=>yes the jagged line is due to our small population which leads to greater variations than for the much larger population of continental France. However, despite the noise there is a strong significant negative trend in French Guiana with a Spearman’s Rho of -0.73, P=0.0009. (the relation is stronger for mainland France Rho=-0.97).
In addition, the year 2017 which is referenced, should also be depicted on the x axis either directly or the time frame should be extended to 2018 as this seems to be the scale here.
=>ok thank you we have modified the x axis
In 1B the increase in French Guiana is not in 2017 but rather from 2017. There is still some time until the pandemic from this point and an explanation for the rise in this timeframe 2018 -2020 is missing. The blockade of major roads and the health care strike may provide explanation for 2017 but not for the following years per se. Please add.
=>ok we have reformulated more carefully. The 3-year scale allows to reduce yearly fluctuations mentioned above but they make it more difficult to pinpoint events.
In 1C there is an up and down almost undulating pattern starting around 2009 in French Guiana that should be subject for further investigation and discussion but it isn’t addressed. Further, the Figure 1 description in L105 uses lower case but the figure itself has capital letters. Please adjust.
=>Ok thank you. For 1C it is deaths over 64 years of age where fluctuations linked to small size are even more pronounced because there are only 15,000 persons over 64 years in French Guiana so, with such a denominator, a few more deaths may lead to fluctuations of rates. We explained more and adjusted.
- L107: It is unclear why in the under 5 mortality 3.3 other South American nations were included but not in 3.2 – 3.9 except for 3.9.3 where Brazil is referenced. Please add/explain.
=> French Guiana is not a country therefore it is mostly absent from international data sources. We use the official French national statistics for the country and regions to describe health indicators. we would have really loved to obtain comparable data for 3.2-3.9 but unfortunately when looking at data sources https://vizhub.healthdata.org the data seemed to have undergone transformations that led to comparability issues. (for example when comparing France data in this source the standardized mortality is much lower than in the official French statistics and we felt that using a rule of 3 to transform the data was crude and a bit of a black box therefore we refrained for going any further).
- L107-135: References to figures should be in cadence in the text and addressed in order. Please adjust and present the results in order of the figures. First Fig1 then Fig2 and so on.
=>ok thank you, we have shifted the sections discussing the Figures called in the text to the level of the figure.
- L136 - 137: Why was this tree map not done for mainland France as a comparison as this is what the authors state in the title?
=> ok we have added it
- L154 - 159: add explanation for why data points intersect with mainland France in certain years.
=>the temporal trends for all indicators in French Guiana is always very jagged because with a population of circa 300 000, a few more cases in a given year may be very visible whereas in France (population circa 60000000) temporal trends look much smoother. The reason is that small samples often have more extreme values. We have added explanations in the limitations.
- L166 - 167: First diagram upper left: What does the description of the y axis mean? Please explain or adjust.
=> thank you we added y axis titles
- L174: Explain why low vaccine uptake of mothers leads to increased infant death and if so why infant death was lower in vaccinated mothers as the statement suggests but does not address.
=>we do not mean in mothers and infant deaths. French Guiana has the lower vaccination rate in Latin america and France which led to many adult deaths, notably young adult deaths which weighed on life expectancy at birth. The subheading at birth is misleading because it does not explicitly mention that we are talking about life expectancy. We rephrased to avoid ambiguity.
- Figure 7: It is unclear what exactly the y-axis numbers represent in each graph. E.g. It seems impossible that life expectancy at 60 years is between 18 and 28. Please correct or explain.
=>the y axis represents the number of years one could expect to live at 20, at 40 and at 60 years. For the example above this would mean that at 60 persons could expect to live until 78 and 88 years.
- Line 178: There is no a,b,c,or d in the figure.
=>thank you we corrected
- Line 194: Exchange “this” to “its”
=>ok thank you
- Line 212: Exchange “with” to “which” add “a” before “role”
=>ok thank you
- Line 213 - 214: Why is that so? There is no explanation or a reference that supports that statement. Add reference or explain.
=>the women from eastern Suriname mostly cross the river to deliver at the sole hospital in the region and they also hope that for their child birth in a French territory will open rights. We have added a reference describing the activation of a plan blanc (emergency mobilizations of all physicians in the face of a crisis, here the influx of pregnant women).
- Line 228: Is this an assumption or a fact? There is no reference for that statement. Please add reference if factual or use “may have further increased…”.
=>ok we rephrased
- Lines 241ff: Also discuss environmental factors such as air quality that may contribute to the fact.
=>ok indeed air quality is much better in the Amazon. we added this
- Lines 255 and 256: change “a dose” to “one dose”, respectively.
=>Ok thank you
- Line 257: Add space after reference [20].
=>OK
- Line 274 – 275: Such demographic and ethnic differences to mainland France should be discussed in more detail. It may even be indicated to look at different ethnicities separately in order to make better conclusions about the health status of one group over another. Please add.
=>We wholeheartedly agree in principle but ethnicity is a touchy topic in France and usually we do not have access to it. Echoes of the second world war and fear of ethnic statistics leading to discrimination explain this. When we mention maroons it is a well known fact in French Guiana but for other problems there is little data, often the proxy is nationality, or language. However, the data presented here does not go to this level of detail even regarding nationality it is aggregated data. We discussed this a bit more.
- Line 302: add “in” after “decline”
=>thank you for spotting this
- Line 304: change “country” to “countries”
=>OK we rephrased
- Line 308: A direct comparison is not indicated. There is probably a difference between living under the French poverty level in mainland France vs French Guiana. This difference should be discussed and if possible the data should be normalized in a meaningful way so that the data can be compared properly.
=>The French poverty level is the poverty level in French Guiana we are not comparing to France. We rephrased to remove the ambiguity of the term French poverty level. Thank you for pointing this.
- Line 324: Move the full stop behind reference [23].
=>OK
- Line 329 – 330: The issue of fake news has not been addressed throughout the manuscript (see point above). So why is it mentioned at the end of the discussion? The point is valid but please add data to support it. I suggest a major revision addressing the raised points.
=>indeed the fake news jumps at the reader during the conclusion without any prior mention of this. Our goal not being this topic we rephrased and talked about vaccine hesitancy instead which echoes the low vaccination rates mentioned.
Again we are very grateful for having benefitted from the thorough and stimulating review. We hope that you will find the manuscript has sufficiently improved.
Reviewer 2 Report
Comparing French Guinea with the french mainland and other countries in South America is an interesting topic.
Some comments:
1. Line 54: "Thus, 29% of the population and nearly half of the adults 54 are of foreign origin." These percentages are inconsisstent.
2. Figure 2: A logarithmic scale has been used. Is it a base 10 logartihm? Please indicate the base.
3. Standardized death rates: Which stanard population has been used?
4. Figures 1: Mortality rates are only compared between French Guinea and the mainland. It would also be interesting comparing mortality rates with those of other countries in South America.
5. Figure 3: A treemap for the French mainland would enable direct comparisons regarding the causes of premature deaths.
6. Figure 6 (Death from infectious diseases excluding tuberculosis etc.): Starting in 2007/09, the number of deaths in French Guinea is less than in France. Please discuss this phenomenon.
7. Figure 8: There is no legend. A corresponding figure for the French population woulrd be helful.
Author Response
Dear Reviewer, thank you for the suggestions, we are very grateful.
You will find the responses inserted below after each comment
Sincerely
The authors
Comments and Suggestions for Authors
Comparing French Guinea with the french mainland and other countries in South America is an interesting topic.
Some comments:
- Line 54: "Thus, 29% of the population and nearly half of the adults 54 are of foreign origin." These percentages are inconsisstent.
=> in fact this is because the median age is 23 years hence nearly 40% of the population are minors. Since they are born in French Guiana they are French even if they have foreign parents. When counting them 29% are foreigners but when we only count adults the proportion of foreigners is much higher.
- Figure 2: A logarithmic scale has been used. Is it a base 10 logartihm? Please indicate the base.
=>yes thank you it is crucial to specify this: it is a base 10 logarithm scale we have specified
- Standardized death rates: Which stanard population has been used?
=>indeed this was missing we specified it was the 2006 census French population.
- Figures 1: Mortality rates are only compared between French Guinea and the mainland. It would also be interesting comparing mortality rates with those of other countries in South America.
=>yes we would have loved that but were not able to obtain this data for other countries. Some websites provide very nice burden of disease data but it concerns the whole population but there is no information on mortality in all age groups. Ourworldindata provides data (but rarely for French Guiana which is pooled with France) but obviously it is standardized in a different way and we were weary to compare different data types so we refrained from doing so.
- Figure 3: A treemap for the French mainland would enable direct comparisons regarding the causes of premature deaths.
=>ok indeed it seems useful for comparisons. We added it and shifted figure numbers after.
- Figure 6 (Death from infectious diseases excluding tuberculosis etc.): Starting in 2007/09, the number of deaths in French Guinea is less than in France. Please discuss this phenomenon.
=>indeed an interesting phenomenon for which we can only speculate. We suspect the broader base of the age pyramid leads to fewer deaths from other infections. We have pointed this in the results and added this in the discussion.
- Figure 8: There is no legend. A corresponding figure for the French population woulrd be helful.
=>Thank you. We have added a legend. An age pyramid for continental France was added as a supplementary document.
Round 2
Reviewer 1 Report
Thank you for addressing my raised points and engaging in a fruitful dicussion that imho and with no doubt has improved the quality of the results and discussion of the presented data even more. Im fully satisfied and accept the author's explanations and addendums. Here are a few minor things I noticed in the revised version that I'd like to bring to your attention. There is no need to get back to me an these points. It is just a heads up.
L118: change 1b to 1B
L125 change 1c to 1C
L183 Figure 7 description is now missing
Author Response
Dear reviewer, again thank you so much for the rapid yet very attentive reading and for the many useful suggestions.
It has been a great process to help us improve the manuscript.
We have corrected as you suggested hereunder.
L118: change 1b to 1B=>OK
L125 change 1c to 1C=>OK
L183 Figure 7 description is now missing=>OK we inserted it